# Manipulating topological transformations of polar structures through real-time observation of the dynamic polarization evolution

K. Du[1,7], M. Zhang [2,7], C. Dai[3,7], Z.N. Zhou[1], Y.W. Xie [2], Z.H. Ren[4], H. Tian [1,4*], L.Q. Chen[3], Gustaaf Van Tendeloo[5,6] & Z. Zhang[1,4*]

Topological structures based on controllable ferroelectric or ferromagnetic domain configurations offer the opportunity to develop microelectronic devices such as high-density memories. Despite the increasing experimental and theoretical insights into various domain structures (such as polar spirals, polar wave, polar vortex) over the past decade, manipulating the topological transformations of polar structures and comprehensively understanding its underlying mechanism remains lacking. By conducting an in-situ non-contact bias technique, here we systematically investigate the real-time topological transformations of polar structures in PbTiO$_3$/SrTiO$_3$ multilayers at an atomic level. The procedure of vortex pair splitting and the transformation from polar vortex to polar wave and out-of-plane polarization are observed step by step. Furthermore, the redistribution of charge in various topological structures has been demonstrated under an external bias. This provides new insights for the symbiosis of polar and charge and offers an opportunity for a new generation of microelectronic devices.

[1] Center of Electron Microscopy, School of Materials Science and Engineering, Zhejiang University, Hangzhou 310027, China. [2] Department of Physics, Zhejiang University, Hangzhou 310027, China. [3] Department of Materials Science and Engineering, Pennsylvania State University, State College, PA 16802, USA. [4] State Key Laboratory of Silicon Materials, School of Materials Science and Engineering, Zhejiang University, Hangzhou 310027, China. [5] Electron Microscopy for Materials Science (EMAT), University of Antwerp, Groenenborgerlaan 171, B-2020 Antwerp, Belgium. [6] Nanostructure Research Centre (NRC) Wuhan University of Technology, Wuhan 430070, China. [7] These authors contribute equally: K. Du, M. Zhang, C. Dai. *email: hetian@zju.edu.cn; zezhang@zju.edu.cn

Domain structures in materials are of great significance considering their relationship with ferroelectricity, conductivity, magnetism, and other abundant materials properties. A variety of topological polar structures have been discovered and evaluated, such as flux closure and vortex domain structures[1–8]. It has been theoretically predicated that an electric field or a mechanical loading has the ability to switch domain configurations between vortex and other ferroelectric domains in ferroelectric systems[9,10]. Exploring a practical pathway to engineer these topological defects and associative states of matter (e.g., polar spirals and skyrmions[11–13]) is highly desired from both academic importance and promising technological applications. Particularly, the electric-field control of topological structures is of great interest because it offers the potential for new cross-coupled functions[14–16]. More importantly, considering the extremely short periodicity of the vortex domains, with various switchable domain patterns which can be modulated by an external factor, $PbTiO_3/SrTiO_3$ (PTO/STO) multilayers may pave the way for the implementation and design of a new generation of high-density microelectronic devices based on domain engineering.

However, existing reports focus on macro scale and ex-situ manifestation, direct observation of the dynamic evolution of a phase transition at an atomic level remains a great challenge. Using a combination of scanning transmission electron microscopy (STEM) and in-situ non-contact bias technique, a real time mapping of the ferroelectric polarization under an electric bias is carried out on an atomic scale. The electron energy loss spectroscopy (EELS) results provide hints on the variation of the electronic structure during the topological transformations. Our methods allow us to realize a controllable transformation, and the connection of various topological structures is experimentally presented. The results are expected to not only shed light on the nature of formation and evolution of topological polar structures

but also provide a general platform for domain engineering in ferroelectrics, which is an important step towards their practical applications in data storage.

## Results

**The variation of polarization distributions in multilayer.** We designed and grew PTO/STO multilayer films on a $SrRuO_3$/$DyScO_3$ substrate. $PTO_{(n)}/STO_{(10)}$ multilayers with $n = 1-21$ (the gradient of thickness is around 2 unit cells) were grown by pulsed laser deposition (PLD) and characterized by Cs-corrected STEM. A typical low-magnification high-angle annular dark-field STEM (HAADF-STEM) image of the cross-section reveals the layer uniformity (Fig. 1a). Atomic resolution EDS-mapping confirms sharp and coherent interfaces (Fig. 1b and Supplementary Fig. 1). Implementing a displacement algorithm, the vector map of the polar displacements within the $n = 1-21$ multilayer (Fig. 1c) shows the domain evolution process. With the thickness of the PTO layer continuously increasing from 2 unit cells (uc) to over 20 uc, the corresponding domain configuration evolves from a/c domain (in which the polarization direction is along the a or c axis) to an unexpected stable wave state and vortex state, and finally to a flux closure state. Electron diffraction and dark-field transmission electron microscopy confirm that these topological structures extend over a long distance (see Supplementary Fig. 2). The variation process within the $PTO_{(n)}/STO_{(10)}$ film is somewhat different from that in the previously reported $PTO_{(n)}/STO_{(n)}$ superlattices[3]. Phase-field modeling of the $PTO_{(n)}/STO_{(10)}$ multilayer was performed to ascertain the evolution process of the domain patterns as a function of the PTO thickness (see Supplementary Fig. 3).

In order to accurately verify the position of all atoms, including oxygen, and to evaluate the spontaneous polarization unit cell by unit cell, the Integrated Differential Phase Contrast (iDPC) STEM

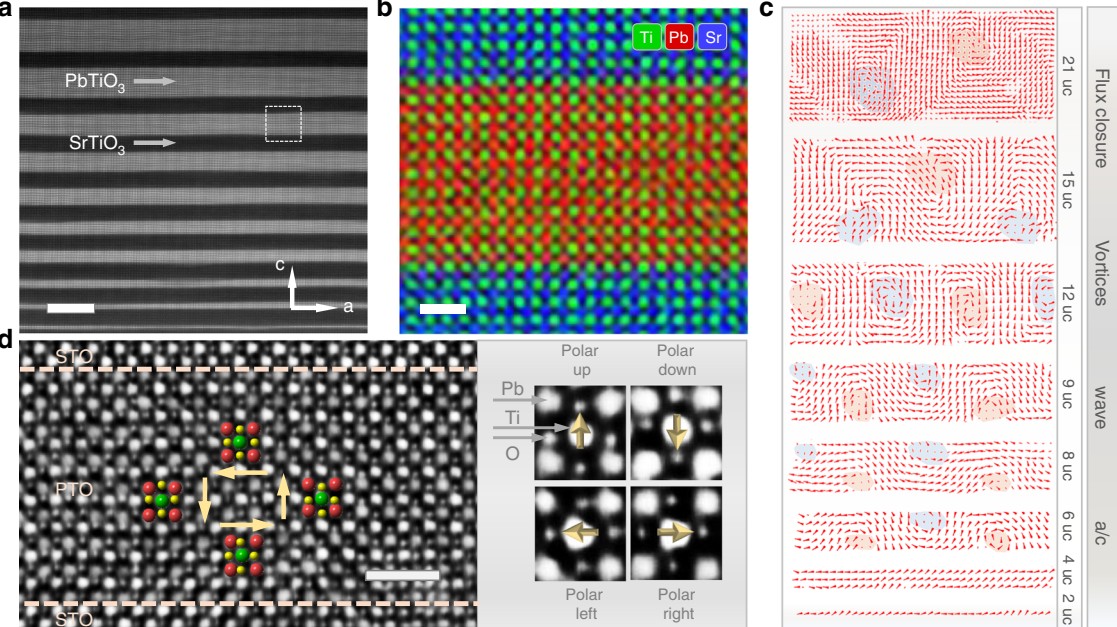

**Fig. 1** Structural characterization of the $PTO_{(n)}/STO_{(10)}$ multilayer. **a** HAADF-STEM image of the film, recorded with the incident electron beam parallel to the $[010]_{pc}$ direction. Scale bar, 10 nm. **b** EDS-mapping of Pb, Sr, and Ti shows that the interface of PTO/STO is atomically sharp. This region is extracted from the white box in **a**. Scale bar, 1 nm. **c** Ti atom displacement vector maps based on experiment, showing the domain evolution with changing PTO thickness. Red and blue regions indicate the clockwise and counterclockwise vortex pairs[59–61]. **d** Superposition of the iDPC-STEM image and a structure model of PTO, showing the atom displacements around the vortex structure. The red, green, and yellow dots denote the positions of $Pb^{2+}$, $Ti^{4+}$, and $O^{2-}$ columns, respectively. Arrows denote the polarization direction. The dashed lines indicate the interface between PTO and STO. The zoom-in images attached to this iDPC-STEM image exhibit the unit cell of PTO along $[010]_{pc}$ in which positions of Ti, O, and Pb are clearly shown. Scale bar, 1 nm

imaging technique[17] is employed (Fig. 1d and Supplementary Fig. 4). Compared with annular bright-field (ABF) STEM, iDPC provides better precision for measuring the atomic column positions[18].

Such domain evolution is usually accompanied by a complicated interaction of multi-factors[3,7,9,10] and therefore the strain evolution needs to be clarified first. The corresponding strain evolution across multilayers from 1 uc to 21 uc is confirmed via the geometric phase analysis (GPA) (see Supplementary Figs. 5–7). These facts point towards a close relationship between the evolution process of the polar structure and the strain evolution, which has been observed earlier[2,19–23]. To explore the physical origin underlying this transformation, we calculated the evolution of the energy components of the ferroelectric domain, as shown in Supplementary Fig. 8; the decrease of the energy density of the PTO layer is accompanied with an increase of the electric and gradient energy density and a drop of the average elastic and Landau energy density. This confirms the phase transition sequence with increasing PTO thickness from the energy aspect, which supports our following experimental observations and gives a theoretical clarification on the domain evolution.

**In-situ real-time mapping of topological transformations.** Although the domain evolution can be experimentally realized by changing the layer thickness, modulating the domain pattern in such a way is obviously unpractical toward application. A more practical way of modulating the domain pattern is needed. Therefore, we developed an approach through the application of an external electric field. The electric field-driven evolution of the vortex was realized under an applied out-of-plane voltage. A 10-uc-thick PTO layer with a vortex structure is selected to perform the in-situ experiment. A bias of 0 ~5 V was applied and real-time observation under atomic resolution was realized by taking an in-situ non-contact bias technique. As the applied voltage is increased, the arrangement of the vortex cores first reveals a zig-zag state, then transforms into a wave state upon reaching the interface of PTO/STO, and eventually evolves into a polarization down state. Phase-field modeling was performed and the results match well with our in-situ bias experiments (see Fig. 2b–d).

An analysis on the time-dependent evolution of the energy components of a ferroelectric domain under different bias was performed (see Supplementary Fig. 9). With applying a mediate electric field, the system prefers a wave-like structure with less out-of-plane polarization, rather than forming a uniformly poled state. Based on phase-field calculations, this is driven by a competition among electric, Landau, gradient and elastic energies as it attempts to reach a new equilibrium. With a higher electric field, the evolution driving force is mainly from the large decrease of electric energy, which overcomes the barrier of Landau and elastic energy (see Methods and Supplementary Fig. 9 for detail).

While for the topological transformations manipulated by an external electric field, the strain under a bias of 2 V and 5 V remains almost unchanged compared with that in the vortex structure at 0 V (see Fig. 2e and details in Supplementary Fig. 9): the out-of-plane strain in the wave structure at 2 V is slightly larger than that in the vortex structure at 0 V (an average 5.1% out-of-plane strain at 2 V, compared with an average 4.0% out-of-plane strain at 0 V), with their pattern unchanged. The zig-zag shape of the strain arrangement presents a little distortion in the polar down state at 5 V. For the in-plane strain, both the strain pattern and strength remain almost unchanged. These results imply that other factors, besides the strain, such as charge might play a more important role during the transformations under electric bias. Exploring the electronic structure is therefore meaningful.

**Exploring the electronic structure of domain configurations.** To explore the electronic structure within these domain

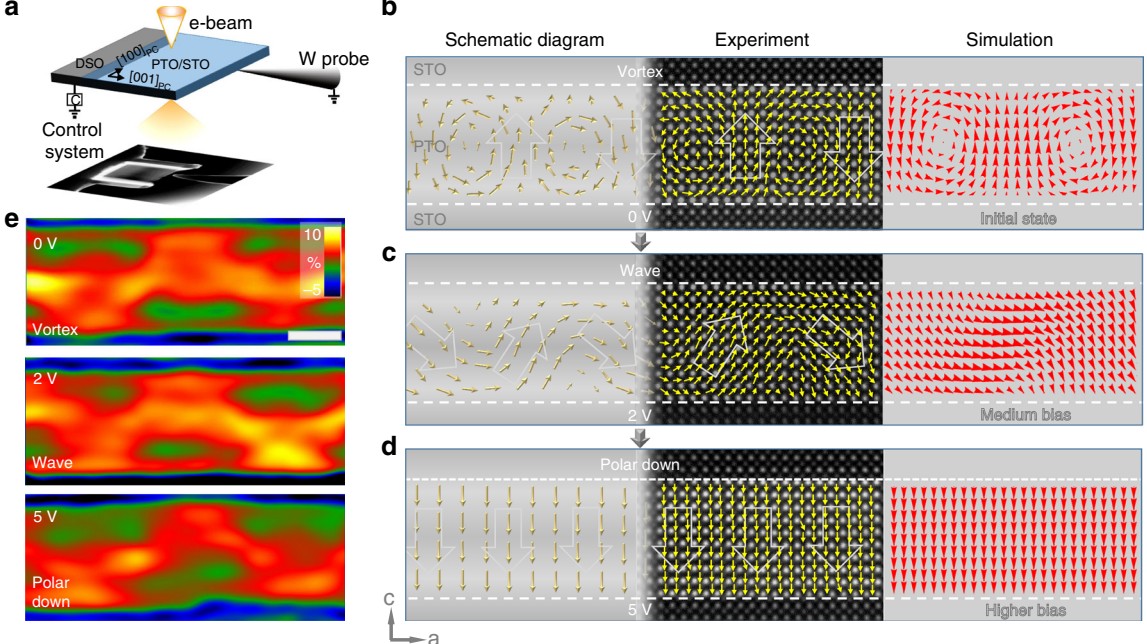

**Fig. 2** Manipulation of the ferroelectric domain in a PTO/STO multilayer. **a** Schematic diagram of the in-situ electric field experiment. The input voltage is applied along the c-axis. To capture the structural changes, a real-time analysis is performed in STEM mode. **b–d** Mapping of the real-time dynamic evolution of the polarization based on HAADF images under an external electric field and the corresponding simulation results. An evolution process of the polar structure from vortex to wave and finally polar down state was recorded. The three columnar sub-panels are schematic diagrams, experimental mappings of polar vector and phase-field simulations, respectively. **e** The out-of-plane strain ($\varepsilon_{yy}$) map of the PTO layer under different external bias, which remain almost unchanged. Scale bar, 2 nm

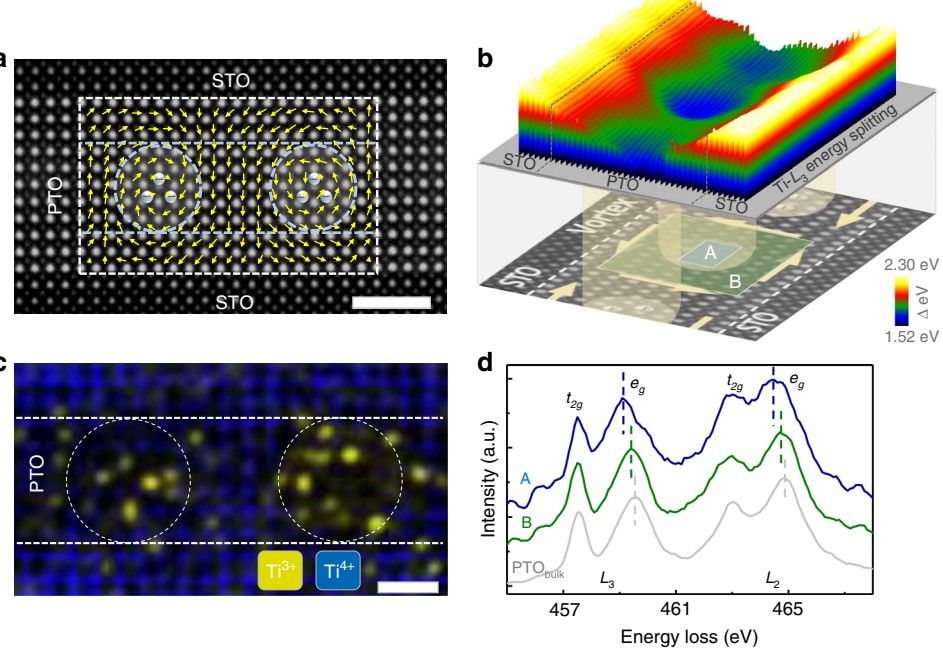

**Fig. 3** EELS-mapping results in PTO$_{(11)}$/STO$_{(10)}$ where vortex domains exist. **a** HAADF image of a PTO layer of 11 uc with vortex domains, together with its polar map. Dashed circles represent the concentrated areas of negative charge. Scale bar, 2 nm. **b** The upper colored surface plot shows the Ti–$L_3$ energy splitting in the PTO layer where a vortex exists, the lower HAADF image shows the location of the corresponding vortex structure in the PTO layer. **c** Superposition of the Ti$^{4+}$ and Ti$^{3+}$ signal based on EELS analysis; the investigated area is extracted from the dashed box area in **a**. Scale bar, 1 nm. **d** Ti–$L_{2,3}$ spectra corresponding to three areas: the blue-A curve is from the vortex core, the green-B curve is from the PTO layer but away from the vortex cores, and the gray curve is the reference Ti–$L_{2,3}$ spectrum acquired in the bulk-PTO. These three areas show a distinct variation of the EELS spectrum, which is a clear indication of the valence changes of Ti

configurations, atomic scale STEM-EELS mapping was performed (see Fig. 3). This is based on a common method, using the valence change of the cations to evaluate the type and relative density of the charge[24,25] or reveal the electron accumulation[26]. We acquired 2D core loss spectral images from a PTO layer with vortex structure. Using model-based quantification of the EELS spectra[27], the relative concentrations of Ti$^{4+}$ and Ti$^{3+}$ were estimated and a clear hint of electron concentration at the core of vortex was found (the concentrated areas of Ti$^{3+}$ are shown in Fig. 3c). According to the model-based quantification, the average Ti$^{3+}$ fraction in a typical core-region is calculated to be ~0.19. As a cross-check of the model-based quantification of the Ti valence, we further measured the $t_{2g}$–$e_g$ splitting of Ti–$L_{2,3}$ (Fig. 3b, d). According to literature[28–31], the splitting between $t_{2g}$ and $e_g$ can be used to measure the proportion of the Ti$^{4+}$ and Ti$^{3+}$ components. A typical Ti$^{3+}$ EELS spectrum of Ti$_2$O$_3$ has two peaks of $L_3$ and $L_2$ edges[32] while there are four peaks for the Ti$^{4+}$ spectrum[33]; correspondingly a narrow split stands for a larger proportion of the Ti$^{3+}$ component[32–34]. On the basis of our measurements, the energy splitting of the $L_3$ and $L_2$ edge at the vortex core is found to be lower than that in other regions of the PTO layer, as shown in Fig. 3d. Moreover, when comparing the octahedral deformations (such as octahedral elongation/tilt) in the vortex core and in non-core regions, no significant difference was observed (Supplementary Fig. 10), indicating that octahedral deformation is not the main reason of such huge $t_{2g}$–$e_g$ splitting change at the core of the vortex. Based on these experimental evidence, the possibility of a vortex core containing a higher Ti$^{3+}$ concentration is reasonably proposed.

An indication of oxygen vacancies at the core of vortex and nearby region was also observed (Supplementary Fig. 11). The presence of oxygen vacancies can provide electrons, which supports our findings of Ti$^{3+}$. A charge accumulation leading

to valance changing has been found in many ferroelectric sysytem[24–26]. For example, the head to head charged domain wall in BFO, where electrons accumulate, is n-type conductive and associated with the presence of Fe valence dropping[26]. Similar features were found in our case, the concentration of electrons may partially occupy Ti 3d states leading to a valence change from Ti$^{4+}$ to Ti$^{3+}$ at the ferroelectric vortex cores. We therefore reasonably expected a concentration of electrons leading to an enhanced electric conductivity at the ferroelectric vortex cores; a similar phenomenon was reported in BiFeO$_3$[35].

To unambiguously demonstrate the correspondence between vortex core and Ti$^{3+}$ component distribution, we analyzed those PTO layers in which the core of the vortex reveals a zig-zag state. The 2D core loss spectra shows that the distribution of Ti$^{3+}$ also reveals a zig-zag pattern (see Supplementary Fig. 12). This correspondence between core and Ti$^{3+}$ can also be found in the polar wave structure: the Ti$^{3+}$ concentrated regions are found to locate at the opening of the wave domain. As a result, the $e_g$ and $t_{2g}$ energy splitting of Ti–$L_3$ shows a clear decrease at these regions (see Fig. 4a, b).

The close relationship between domain and Ti$^{3+}$ component distribution was also found in the out-of-plane polarization region. There are some out-of-plane polarization regions which usually exist in a/c domains and some wave structures (for example, the wave domain shown in Supplementary Fig. 5e). According to a previous study[36], for a single-domain ferroelectric perovskite oxide, a high concentration of electrons (with bulk density of the order of ~$10^{20}$ cm$^{-3}$)[37] or oxygen vacancies accumulated near the polar surface is needed for screening. EELS investigations show that oxygen vacancies are accumulated at the negative polar interface[38,39] as shown in Fig. 4c, d. Moreover, a closer Ti $t_{2g}$–$e_g$ split was found as shown in Fig. 4e, indicating the existence of Ti$^{3+}$ near the positive polar interface, and a

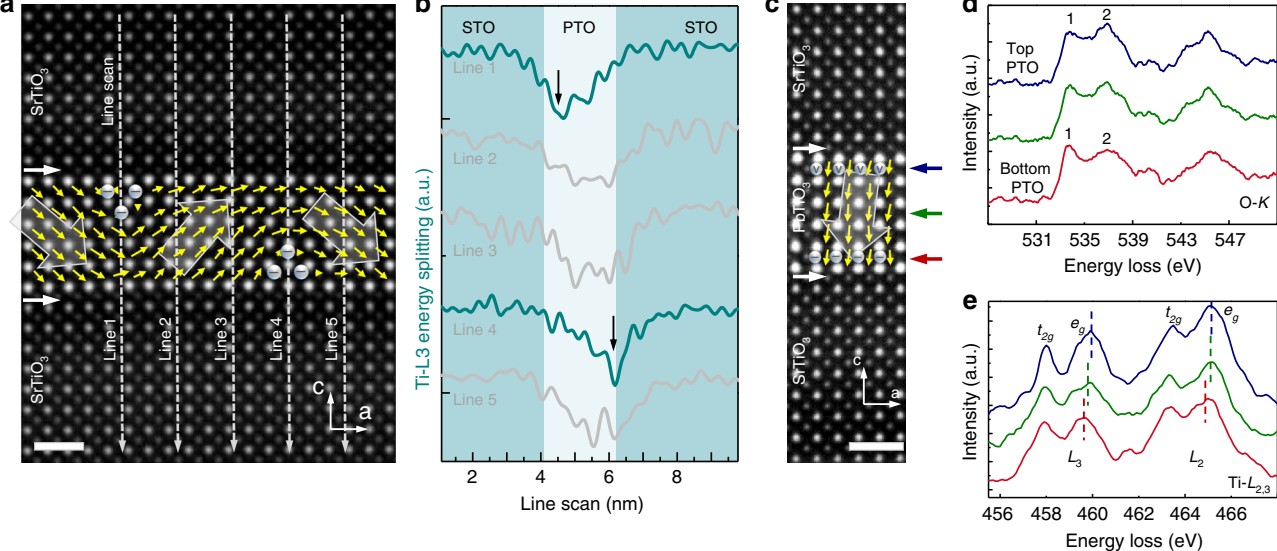

**Fig. 4** HAADF-STEM images and EELS characterization across the multilayer. **a** HAADF-STEM image of PTO$_{(6)}$/STO$_{(10)}$. The yellow arrows indicate the polarization direction. Line scan EELS were performed across different regions of the wave domain. Scale bar, 1 nm. **b** Energy splitting value in the PTO and STO layers across the corresponding regions in **a**, showing that the Ti$^{3+}$ component is constrained in the wave opening. **c–e** HAADF-STEM image of the polar down region and corresponding EELS results of Ti–$L_{2,3}$ and O–$K$ edges across the PTO layer. The blue, green and red curve are acquired at the regions indicated by arrows in **c**. The reference O–$K$ spectrum acquired in PTO bulk could be seen in Supplementary Fig. 11. Subtle changes are observed as a result of an electronic-structure reconstruction. These results provide evidence for the appearance of oxygen vacancies on the negative polar interface of PTO and Ti$^{3+}$ component on the positive polar interface. Scale bar, 1 nm

continuous increase of the Ti$^{3+}$/Ti$^{4+}$ ratio across the out-of-plane polarization region, obtained by model-based quantification of the EELS spectra, also shows good agreement with energy splitting analysis (see Supplementary Fig. 13). These experimental evidence above point toward the idea that the polarization screening within the out-of-plane polarization regions is realized by the existence of oxygen vacancies on the negative polar interface and the accumulation of electrons on the positive polar interface. This is consistent with the screening mechanism which has been well established in similar ferroelectric materials[24,25,40].

## Discussion

Previous researches have studied various topological structures[1,2,7], and their corresponding strain mappings differ from one another in terms of magnitude and pattern. However, in our in-situ real-time atomic observation on topological transformation under the external electric field, a dynamic rotational switching process is observed, which is very similar to theoretical predictions[41] but with no distinct variation on the strain. Also, note that the motion process of the vortex cores under the electric field (see Supplementary Fig. 14) generally matches previous theoretical results[42]. Specifically, the two vortex lines first move toward each other to reduce the area with opposite polarization directions. However, instead of "melting"[42], the vortex lines move away from each other again under a higher bias.

We analyzed a series of atomic EELS mappings on different polar structures; the presence of Ti$^{3+}$ and oxygen vacancies provides a cross-check that shows a high correlation with the various topological structures. These experimental evidence allow us to propose the possibility of an one-to-one correlation between the charge and various domain configurations in this oxide multilayer. Our findings support that in the out-of-plane polarization layer, the electrons concentrated at the positive polar interface may be attributed to the polar screening. While in the vortex, zig-zag vortex state, and the polar wave structure, where polar screening is not needed, other factors such as the gradient of

the electric field or the local pressure may make a contribution. An analogous phenomenon was found and explained in the field of optical vortices[43,44]: optical tweezers are able to trap and manipulate small particles, typically of micron size. Another possible origination is that the formation of oxygen vacancies at the vortex cores would attract electrons. The local pressure at the vortex cores is negative, thus oxygen vacancies are favored to distribute near the core-region, because oxygen vacancies have larger atomic volume than oxygen atoms. However, a full and in-depth understanding of the underlying mechanism still requires further research.

In summary, we successfully developed an in-situ non-contact bias technique approach to modulate topological polar structures, and for the first time, we demonstrate a straightforward view of a real-time atomistic evolution process in topological transformations. The EELS analyses explored the accompanying changes in electronic structure. Our experimental data clearly suggest a meaningful method to induce transformations among various ferroelectric domains, providing new insights into the formation and stability of various domain structures.

## Methods

**Material system.** The PbTiO$_{3(n)}$/SrTiO$_{3(10)}$ multilayers were epitaxially grown on orthorhombic DyScO$_3$(110) substrates with a ~5 nm SrRuO$_3$ buffer layer using pulsed laser deposition. To obtain well-defined ScO$_{2-}$ terminated substrates, the DyScO$_3$ substrates were treated before deposition as follows. They were first annealed at 1000 °C for 2 hours in air, and then wet etched using 12 and 1 M NaOH-DI water solution in an ultrasonic bath for 1 h[45]. While SrTiO$_3$ and SrRuO$_3$ targets were stoichiometric, a Pb$_{1.2}$TiO$_3$ target was used to ensure layer-by-layer growth. All depositions were performed at 600 °C with an oxygen pressure of 0.1 mbar. A KrF laser (248 nm) was used with a repetition rate of 2 Hz. The laser fluence was 2.1 J/cm$^2$ for depositing the SrRuO$_3$ buffer layer, and 1.5 J/cm$^2$ for depositing the SrTiO$_3$/PbTiO$_3$ multilayer. After that, the thickness of the SrTiO$_3$ films was fixed at 10 uc and PbTiO$_3$ films varied gradually from 1 uc to 21 uc and from 5 uc to 11 uc. The layer-by-layer mode was maintained throughout the entire synthetic process. The thickness of each film was monitored by Reflection High Energy Electron Diffraction (RHEED). After deposition, the samples were cooled down to room temperature under an oxygen partial pressure of 200 mbar to promote full oxidization.

**Conventional and scanning transmission electron microscopy**. Samples were cut into lamellas with the widest faces perpendicular to the [010] direction using Focused Ion Beam (FEI Quanta 3D FEG) for observation by transmission electron microscopy. We used spherical aberration corrected electron microscopy (FEI Titan G2 80–200 ChemiSTEM, 30 mrad convergence angle, 0.8 Å spatial resolution) to acquire atomic resolution HAADF-STEM images; the image noise was corrected using Digital Micrograph. All STEM images in this work are filtered in Fourier space using a grid mask to select for the lattice frequencies and by low and high pass annular filters to remove the zero frequency and high frequency noise above the information transfer limit. Electron Energy Loss Spectroscopy (EELS) was also performed to verify whether there are changes in the valence state by using a Themis G2 60–300 at 120 kV (with EELS energy resolution ≤0.3 eV). As the bright dots of every atom column in a HAADF image are not always very symmetrical, we determine the center of the atom column by a mathematical method involving Gaussian Fitting based on Matlab. Polarization mapping is then performed by calculating ion displacements in the HAADF-STEM images. SAED (selected area electron diffraction) obtained on a FEI Tecnai G2 F20 S-TWIN is used to verify the analysis of the local evolution in a statistical sense. As the local atom displacements are extremely tiny, most HAADF images are overlaid based on 12 or more images to obtain more accurate atom positions and to correct the small sample drift (<0.5 Å min$^{-1}$). In order to minimize the potential damage caused by the ion beam in FIB, we adjusted the voltage down to 2 kV and the electric current of ion beam down to 27 pA.

**GPA analysis**. The strain fields in this article were deduced for all HAADF-STEM images using custom plugins of GPA for Gatan Digital Micrograph[46,47]. The strain in the STO lattice is relatively smaller and the STO layer is usually used as a reference. The lattice parameters of PTO and STO are slightly different, and a normalization process has been carried out. The GPA is an effective approach to determine crystal lattice variations over a larger area. To further prove that the GPA results are credible and show correspondence with the STEM images, we calculated the a/c ratio and compared it with the GPA results as shown in Supplementary Fig. 7; they show a high degree of consistency.

**In-situ study**. The in-situ biasing was performed using a Hysitron PI-95 TEM Picolndenter. Using a function generator, an electrical bias was applied between a tungsten tip, which acts as a mobile electrode, and the conductive SRO bottom electrode, which is connected to ground. The input voltage is applied between the sharp conductive tip and the sample. Every continuous bias lasts for over 100 s and is then removed for the sample to relax; after that, another bias is applied. The atomic-scale spatial resolution of STEM provides a clear picture of the evolution under an external electric field.

**Detection of oxygen vacancies**. According to the previous literature, the peak located at ~537 eV (marked as peak 2 in Fig. 4d) in the O–K edge is closely related to the bonding state between oxygen and the surrounding cations[38]. The red spectrum exhibits a slightly lower peak 2 compared with the peak located at ~533 eV (marked as peak 1), in agreement with the calculated results of PTO without oxygen vacancies[39]. Differently, the blue spectrum exhibits an obviously enhanced peak 2, comparable to peak 1, providing evidence for the appearance of oxygen vacancies[39].

**Phase-field simulations**. In the phase-field modeling of the PbTiO$_3$/SrTiO$_3$ superlattice, the polarization (**P**) evolution is governed by the time-dependent Ginzburg-Landau (TDGL) equation:

$$\frac{\partial P_i(\mathbf{r}, t)}{\partial t} = -L \frac{\delta F}{\delta P_i(\mathbf{r}, t)} \ (i = 1, 2, 3) \tag{1}$$

where L, r, and t represent the kinetic coefficient, spatial position vectors and time, respectively. The free energy F contains the contributions of the Landau, elastic, electrostatic and gradient energy, *i.e.*,

$$F = \int (f_{\text{Landau}} + f_{\text{Elastic}} + f_{\text{Electric}} + f_{\text{Gradient}}) dV \tag{2}$$

The Landau energy density is expanded by "Landau polynomial".

$$f_{\text{Landau}} \ \alpha_{ij} P_i P_j + \beta_{ijkl} P_i P_j P_k P_l + \gamma_{ijklmn} P_i P_j P_k P_l P_m P_n \tag{3}$$

The elastic energy density can be expressed by the following equations:

$$f_{\text{Elastic}} \ \frac{1}{2} C_{ijkl} e_{ij} e_{kl} = \frac{1}{2} C_{ijkl} (\varepsilon_{ij} - \varepsilon_{ij}^0)(\varepsilon_{kl} - \varepsilon_{kl}^0) \tag{4}$$

where $C_{ijkl}$, $e_{ij}$, $\varepsilon_{ij}$ and $\varepsilon_{ij}^0$ are elastic stiffness tensor, elastic strain, total strain and eigen strain, respectively. The eigen strain is the phase transformation strain given by electromechanical coupling with the expression: $\varepsilon_{ij}^0 = Q_{ijkl} P_k P_l$. $Q_{ijkl}$ is the electrostrictive coefficient tensor.

The electric energy density is calculated as

$$f_{\text{Electric}} = -\frac{1}{2} K_{ij} \varepsilon_0 E_i E_j - E_i P_i \tag{5}$$

where $K_{ij}$ is the background dielectric tensor constant, $\varepsilon_0$ is the dielectric permittivity of the vacuum and $E_i$ is the local electric field defined by possion equation: $E_i = -\nabla_i \varphi$.

With the assumption of pesduocubic, the gradient energy density is given by:

$$f_{\text{Gradient}} \ \frac{1}{2} G_{ijkl} P_{i,j} P_{k,l} \tag{6}$$

where $G_{ijkl}$ is the gradient coefficient tensor, $P_{i,j}$ is for the spatial differential of the polarization vector: $P_{i,j} = \frac{\partial P_i}{\partial X_j}$.

More details of each energy density have been reported previously[48–53]. A three-dimensional system is set up, using a size of $(200\Delta x) \times (200\Delta y) \times (250\Delta z)$ with $\Delta x = \Delta y = \Delta z = 0.4$ nm. The thickness of the substrate, film and air are $30\Delta z$, $141\Delta z$ and $79\Delta z$, respectively. In the film area, 9 unit cells of PbTiO$_3$ layers and 9 unit cells of SrTiO$_3$ ones as well as one transition layer between PbTiO$_3$ and SrTiO$_3$ are used to simulate the superlattice system. The pseudo-cubic lattice constants for PbTiO$_3$ and SrTiO$_3$ are 3.9547 Å and 3.905 Å, respectively. The DyScO$_3$ substrate lattice constants along the x- and y-direction are taken as 3.952 Å and 3.947 Å, respectively[54]. Periodic boundary conditions are applied along the x- and y-directions, and a superposition method is used along the out-of-plane direction[55]. The short-circuit electric condition is applied, where on the top and bottom, electrical potentials are the applied bias and 0, respectively. A mixed elastic boundary condition is assumed where the displacement is zero far away from the substrate, and out-of-plane stress is free at the top of the film[49–52,56]. Random noise is used to simulate the annealing process as the initial nuclei.

**Determination and mapping of the polar atomic displacements**. The polar vector determination was performed on the Cs-corrected HAADF-STEM images using local A- and B-site sublattice offset measurements. In a tetragonal PTO structure at room temperature, Ti$^{4+}$ shifts relative to the Pb sublattice, while oxygen octahedra shift in the same direction but with a larger displacement, as shown in Supplementary Fig. 4; $\delta_O$ and $\delta_{Ti}$ are the displacements along the c-axis. The shift breaks the cubic symmetry, resulting in an internal dipole with a direction pointing from the center of negative charge to the center of positive charge, hence the polarization direction of PTO is opposite to the shift direction of Ti$^{4+}$ and this offset can be used to infer the full polarization (P). Thus, for traditional HAADF images, we use the Ti to infer the offset of the Ti-'centered' oxygen octahedra for defining the electric dipole. Displacement vectors corresponding to local offsets between the A- and B-site sublattices were calculated by determining atomic positions by fitting each atom site by a spherical Gaussian using an algorithm in Matlab[57,58]. For the iDPC-STEM images, the oxygen atoms can also be acquired, so we calculated the spontaneous polarization according to the position of both Ti and O atoms.

## Data availability

The datasets generated during the current study are available from the corresponding author on a reasonable request.

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

## Acknowledgements

This work was supported by the National 973 Program of China (2015CB654901), National Key R&D Program of China (Grant No. 2016YFA0300204, 2017YFB0703100 and 2017YFA0303002) and the 111 Project (Grant No. B16042). We thank Prof. Fang Lin for providing guidance on calculating atoms position. The work at Penn State is supported by the U.S. Department of Energy, Office of Science, Office of Basic Energy Sciences, Division of Materials Sciences and Engineering under Award DE-SC-0012375 and by the National Science Foundation under Grant No.DMR-1744213.

## Author contributions

H.T. and Z.Z. co-designed the project. K.D. performed the experiments related to Electron Microscopy. K.D. took charge of the data analysis. Z.N.Z. designed the displacement vector-mapping algorithm. C.D and L.Q.C did the phase-field modeling. M.Z. and Y.W.X. synthesized the samples. K.D., L.Q.C., Z.H.R., Y.W.X., H.T., and G.V.T. co-write the paper. All authors contributed to the discussions and paper preparation.

## Competing interests

The authors declare no competing interests.
