## [Peer Review File · Nature Communications]

Reviewers' comments:

Reviewer #1 (Remarks to the Author):

The authors report on various ways of controlling polar vortex patterns in PTO/STO heterostructures. The data is well presented and discussed mostly appropriately. While the control through PTO thickness has already been discussed by other groups for this material heterosystem, the electric field control of the vortices and associated changes in chemical structure (EELS) in my opinion make this an interesting and original study. For this reason I can recommend it for publication.

Reviewer #2 (Remarks to the Author):

The work by K. Du et al. reports on a thorough structural, electronic characterization of the PTO/STO superlattices using various STEM techniques and the home-developed polarization evaluation software. All these experimental tasks are challenging and respectable. This report readily deserves an active consideration of the journal from a technical viewpoint. However, similar domain patterns to the rich ones hereby appeared before. What is new about this manuscript is then the in-situ characterization on the domain evolution under an electric-field stimulus. This materials-physics part and related spectroscopic interpretations are nonetheless unsatisfactory in this referee's opinions (given below), affecting the readability and value of the work.

1. In ferroelectrics, the domain formation and corresponding topology are the cooperative effect of the primary and secondary order parameters (spontaneous polarization and ferroelastic strain, respectively, in the pristine PTO of central interest herein). The associated depolarization, strain, and domain-wall energies, therefore, dictate the domain patterns. While the strain effect was mentioned in the main text and SI (see the following point), the electric depolarization and domain wall, both of which are important for the EELS discussion, have not been sufficiently discussed in the main text. Despite the relevant phase-field simulations, the work is still not straightforward to read and shows room of improvements along this line.

2. The strain argument is partly based on the GPA analysis of the HAADF images. Intriguingly, the derived in-plane strain (ϵ_{xx} , Figs. S5) is non-uniform and the out-of-plane counterpart (also Figs. S6 and S7) shows periodic oscillations with a periodicity large than the lattice dimension. Such structural characteristics shall lead to corresponding superlattice spots (though possibly diffused), while the SAED pattern in Fig. S2 does not reveal any relevant hint. How would the authors reconcile this puzzle? Moreover, the GPA method demands for a reference lattice. What was the reference-lattice frame exploited and was it statistically meaningful enough? Is the non-uniform in-plane strain that implies a lifting of the generally applicable pseudomorphic strain real or rather an artifact? The physical origin underlining the topological transformation of the domain patterns as a function of the PTO thickness is not clear to this referee.

3. The appearance of electron carriers in ferroelectrics is fundamentally for screening the diverging electrostatic potential of relevant head-to-head dipoles. By contrast, the vortex core and the polar wave are formed by head-to-tail dipoles as depicted by the authors in Fig. 2, for which neither the elegant dipole closure of a vortex (Fig. 1d) nor the head-to-tail polar-wave intersection (Fig. 4) would require screening charges. As a result, the EELS interpretation of electron accumulations at the vortex core (Fig. 3) and the polar-wave junction (Fig. 4) becomes difficult to understand for this referee. As stated by the authors, the crystal-field splitting in the PTO is different from that in the STO. The direct correlation of a change in the crystal-field splitting to the presence of Ti^{3+} (thus electrons) in Fig. 3d is, therefore, unconvincing to this referee. The change in a crystal-field splitting and the presence of Ti^{3+} component can by far be also evident in the corresponding O K-edge spectra. Did the authors cross-check the Ti L- and O K-edge spectra? Furthermore, did the authors acquire the reference Ti L- and O K-edge spectra in both the bulk-PTO and -STO standards and compare them to the film

counterparts? Is the conclusion drawn from Figs. 3 and 4 still robust? A quantitative estimation of a diverging potential and the screening charges required would be helpful for interpreting the EELS results.

4. STEM-EELS is indeed capable of probing screening charges to ferroelectric dipoles but unable for tackling the bound charges of given dipoles. In this regard, the authors' interpretation of unveiling the ferroelectric-bound charges in a down-pointing film region (Figs. 4c-e) becomes puzzling. Again, comparisons to the bulk-PTO spectra are indispensable in order to clarify the issue.

In summary, the structural characterization part of this work is decent. In comparison, the electronic interpretation part is unsatisfactory and, in some places, contradicting to the established wisdom. Indeed, this electronic part is essential for the titled subject and extensive revisions would be required before further considerations of this manuscript.

Minor point:

The ABF imaging does not require extremely thin specimens according to this referee's hand-on experience. Therefore, it would be important for the authors to be specific about the specimen-thickness requirements of the respective iDPC and ABF techniques for the merit of potential interest parties.

Reviewer #3 (Remarks to the Author):

The manuscript reports atomic scale structure and evolution of topological textures in a PbTiO₃/SrTiO₃ (PTO/STO) superlattice. By using the advanced TEM techniques such as iDPC-STEM, EELS, and in-situ non-contact bias, as well as high-resolution STEM, the authors successfully reveal the delicate ferroelectric structures emerging in the thickness-controlled PTO sub-layers of the superlattice. The image quality looks excellent and convincing. One of the main findings is observation of negative charge accumulation at the core of the topological vortices and the EELS measurement results provide useful hints on the electronic structures at the local areas. Since the topological textures in ferroelectrics are an emerging field attracting a lot of attentions recently, the findings in a hot topological system have potential merits to be published. But, the paper still has room for improvement and raises the following questions and concerns as listed.

- 1) The correct topological terminologies should have used to describe the chiral vortices. Vorticity is a different concept from chirality. In the caption of Fig. 1(c), the phrase of "The red and blue regions indicate the vortex-antivortex pairs." is not correct. According to 2D winding number calculation, two singular points in the red and blue regions appear to have a topological charge of +1, respectively. And thus, they both should be called vortices rather than using the antivortex. Since they can be distinguished in aspect of chirality, "clockwise or counterclockwise chiral vortices" are topologically meaningful expressions to indicate the two regions. More rigorous description of the topological charges in ferroelectrics can be found in recently published references [Kim, K. E. et al. Nature Communications 9, 403 (2018), Li, Y. et al., npj Quantum Materials 2, 43 (2017), Seidel, J. (Ed.) Topological structures in ferroic materials. (Springer, 2016)], which are necessary to be cited in addressing the topological structures.
- 2) Specify where the biased voltage is applied during the in-situ bias experiment. Regardless of the explanation in the Methods section, the experimental geometry of non-contact electric bias experiment is still unclear. Is it possible to quantify the electric field across the corresponding layer of PTO?
- 3) In Figs. 2b-d, the white dots indicating negative charges are displayed on the experimentally obtained polarization maps. Is this charge distribution also experimentally confirmed or just guessed from polarization discontinuity? As in the Fig. 2d, one can easily imagine the bottom interface of

polarization down region has a negative bound charge density. But, it would not seem simple to predict the singularities have negative charges in the vortex and wave configurations. The Figs. 2b-d have three columnar sub-panels consisting of two experiments and one simulation. What does the first column in the experimental part mean?

4) In the following EELS experiments, the t_{2g}-e_g splitting in the L₃ or L₂ peaks is used as an indicator of electron doping. As argued by the authors, the decrease in the splitting can be attributed to the existence of Ti³⁺ along with Ti⁴⁺. But, this may not be the unique reason. Basically, the t_{2g}-e_g level splitting is due to the crystal field from the neighboring anions. The vortex areas have non-collinear arrangements of polarizations, which can result in complex octahedral deformations. Exact electronic structure in the situation is not disclosed yet. Without exclusion of the possible concern or minimal warning, such innocent interpretation can fall into error.

5) How can the authors quantify the ratio of Ti³⁺ and Ti⁴⁺ concentrations in Fig. 4e? Is it proportional to the ratio of integrated intensities of t_{2g} peak relative to the e_g peak because the Ti³⁺ partially occupies the t_{2g} orbitals? If so, the physical meaning of the spectroscopic results should be more kindly and strictly described in the main text related to the figure.

**Point-by-Point Response to Referees**

**Response to Referee #1**

**Comment 1:** The authors report on various ways of controlling polar vortex patterns in
PTO/STO heterostructures. The data is well presented and discussed mostly appropriately. While
the control through PTO thickness has already been discussed by other groups for this material
heterosystem, the electric field control of the vortices and associated changes in chemical
structure (EELS) in my opinion make this an interesting and original study. For this reason I can
recommend it for publication.

**Response:** We'd like to thank the referee for his/her positive evaluation.

We are delighted to see that Referee#1 speaks highly of our work because we demonstrate a
precise way to manipulate various topological. Indeed, a variety of topological polar structures
have been discovered and evaluated before. While different from previous works, we
successfully manipulated the topological structures by in-situ bias and provide direct
visualization of the vortex evolution process at the atomic scale.

Response to Referee #2

General comment: The work by K. Du et al. reports on a thorough structural, electronic characterization of the PTO/STO superlattices using various STEM techniques and the home-developed polarization evaluation software. All these experimental tasks are challenging and respectable. This report readily deserves an active consideration of the journal from a technical viewpoint.

Response: We'd like to thank the referee for his/her positive evaluation on our challenging and subtle experiments. We demonstrated a way to manipulate various topological transformations and a straightforward view of a real-time atomistic evolution process. The connection among different topological structures is experimentally established.

However, similar domain patterns to the rich ones hereby appeared before. What is new about this manuscript is then the in-situ characterization on the domain evolution under an electric-field stimulus. This materials-physics part and related spectroscopic interpretations are nonetheless unsatisfactory in this referee's opinions (given below), affecting the readability and value of the work.

Response: Following the comments/suggestions of the referee, we performed additional experiments and simulations to improve the readability of the manuscript:

- 1) We carried out more EELS analyses, both the O-K and Ti-L fine structure of the spectra are discussed (please see responses to comment 1 & 3 of referee 2). A possible screening mechanism is assumed based on further experimental evidence. Additional analysis on the screening mechanism, depolarization, and domain walls are provided (see main text and responses to comment 1 & 4 of referee 2).
- 2) We measured the a/c ratio and provided a detailed analysis of the diffraction patterns. The energy of a PTO layer as a function of PTO thickness obtained from phase field modeling is also attached here (see response to comment 2 by referee 2). We modified the statement about strain.

We also rewrote the main text to improve the readability of the work. In addition, some terminologies and presentations are clarified.

**Comment 1:** In ferroelectrics, the domain formation and corresponding topology are the
cooperative effect of the primary and secondary order parameters (spontaneous polarization and
ferroelastic strain, respectively, in the pristine PTO of central interest herein). The associated
depolarization, strain, and domain-wall energies, therefore, dictate the domain patterns. While
the strain effect was mentioned in the main text and SI (see the following point), the electric
depolarization and domain wall, both of which are important for the EELS discussion, have not
been sufficiently discussed in the main text. Despite the relevant phase-field simulations, the
work is still not straightforward to read and shows room of improvements along this line.

**Response:** Thanks for the valuable comment from the referee. We agree with the referee that the
vortex formation is influenced by a combination of competing factors such as strain,
depolarization, screening mechanism and electrostatic forces. The vortex structure of PTO/STO
was initially reported by R. Ramesh et.al, (Ref. 1 in the main text), and the competing effects of
strain, depolarization and domain wall energy on the vortex lattice formation are discussed in
Ref. 3. Based on the comment from the referee, we carried out additional EELS experiments on
both Ti-L_{2,3} and O-K edges, and compared the results among different regions (see revised main
text Fig. 3 and Fig. 4 and Fig.R1,R2,R7), in particular:

1) We further studied the strain evolution and revised some of the relevant discussions according
to the referee's suggestions (see response to comment 2 of referee 2, and the supplementary
materials Fig.S5, S6, S7, S8, S9). Furthermore, we demonstrated the topological transformations
under an in-situ bias, in which the strain remains almost unchanged (see Fig. 2e in main text).

2) To understand the role of depolarization and domain walls, we examined three typical areas: i)
out-of-plane polarization regions, ii) the transition region between vertical and horizontal
polarization (like 90° domain-wall), and iii) the vortex core.

i) In the case of out-of-plane polarization regions, EELS investigations show a strong evidence
 that oxygen vacancies are accumulated at the negative polar interface, a Ti 3+ component was
 found at the positive polar interface. This is in good agreement with the polarization screening
 mechanism for depolarization in many ferroelectric interface systems [for example, Ref. 27 and
 Ref. 42]. More explanations and experimental details are present in the response to comment 3 of
 referee 2.

**Fig.R1. High-resolution HAADF-STEM and EELS characterization of a micro region in**
 **the out-of-plane polarization layer. a**, High-resolution HAADF-STEM image of a typical 6uc-
 PTO-layer. **b** and **c**, EELS results of Ti-L2,3 and O-K edges across the PTO layer. The blue,
 green and red curve are acquired at the corresponding regions indicated by arrows in (a).

ii) The polarization rotating region seems like a 90° domain wall, as shown in Fig.R2 (orange-C
 region). The EELS results reveal that the t_{2g} - e_g energy splitting in such regions show no distinct
 variation compared with the vertical or horizontal polarization regions (marked as green), which
 is consistent with common understanding.

iii) We systematically studied the regions where polarization is suppressed that we called “vortex
 core” and included the results in our revised manuscript and the response to comment 3 of
 referee 2. The vector maps of electric and polarization fields in the vortex layer and domain wall

energy have previously been presented in [Nature 565, 468–471 (2019)], and it was shown that
these walls are also the regions where the energy density is larger than in the bulk of the domains.
We included the above discussions in the main text together with Fig. 3.

**Fig. R2. EELS characterization of a micro region in the vortex layer.** **a**, HAADF image
around a vortex region. Three representative regions are marked with different colors. The blue-
A region is located in the vortex core, the green-B region is located in the PTO layer but away
from the vortex cores, the orange-C region located in the transition region between regions with
vertical and horizontal polarization directions. **b**, EELS results of Ti-L_{2,3} edges acquired at
different regions as mentioned in (a), and the grey curve is the reference Ti-L_{2,3} spectra acquired
in the bulk-PTO.

**Comment 2:** The strain argument is partly based on the GPA analysis of the HAADF images.
Intriguingly, the derived in-plane strain (ϵ_{xx} , Figs. S5) is non-uniform and the out-of-plane
counterpart (also Figs. S6 and S7) shows periodic oscillations with a periodicity large than the
lattice dimension. Such structural characteristics shall lead to corresponding superlattice spots
(though possibly diffused), while the SAED pattern in Fig. S2 does not reveal any relevant hint.
How would the authors reconcile this puzzle? Moreover, the GPA method demands for a
reference lattice. What was the reference-lattice frame exploited and was it statistically
meaningful enough? Is the non-uniform in-plane strain that implies a lifting of the generally
applicable pseudomorphic strain real or rather an artifact? The physical origin underlining the

topological transformation of the domain patterns as a function of the PTO thickness is not clear
to this referee.

**Response:** According to our experimental results, not only the out-of-plane strain but also the in-
plane strains exhibit periodicity, while the magnitude of in-plane strain is relatively small thus
the periodicity is not as clearly seen as that for the out-of-plane strain. Following the referee's
suggestion, we added some marker lines in the in-plane strain images to make the periodicity
more visible, and updated the related images in the supplementary material. In the in-plane strain
shown in Fig.S5 as an example, the updated version (shown in Fig.R3c) exhibits a sinusoidal
array, which is very similar to the results of Y. L. Tang et al. (Ref. 2) and Wang, W. Y et al. (Ref.
6).

As the referee correctly argued, such structural characteristics should indeed lead to superlattice
spots. A close inspection of the SAED patterns by magnifying the reflections (see Fig.R3b inset)
indeed reveals additional spots which manifest the long-range ordering of the topological
structures. We have updated the Supplementary Fig. S2.

Regarding the referee's suggestions on the information about the GPA method, Fig. R4 is shown
to present the reference lattice. Large scale strain fields in this article were deduced for all
HAADF-STEM images. The strain in the STO lattice is relatively small, and the STO layer is
usually used as reference. The lattice parameters of PTO and STO are a slightly different, and a
normalization process has been carried out. The GPA is an effective approach to determine
crystal lattice variations over a large area. To further prove that the GPA results are credible and
show correspondence with the STEM images, we calculated the a/c ratio and compared it with
the GPA analysis results as shown in Fig. R4. The a/c ratio and the GPA results show a high
degree of consistency [and they match the results of Y. L. Tang et al (Ref. 2)]. The magnitude of
the strain also exhibits statistically meaningful regularity within the thickness-changed-layer as
shown in Fig. S6.

**Fig. R3. Structural characterization of $(\text{PbTiO}_3)_n/(\text{SrTiO}_3)_{10}$.** **a**, Low-magnification STEM
 cross-section image of a $(\text{PbTiO}_3)_n/(\text{SrTiO}_3)_{10}$ multilayer, with $n=1-21$. **b**, Cross-section DF-
 TEM image of a $(\text{PbTiO}_3)_n/(\text{SrTiO}_3)_{10}$. Each bright/dark modulation corresponds to a period of
 the clockwise-counterclockwise vortices structure. In the SAED pattern, the g -vector of the two-
 beam imaging condition is indicated by a yellow circle. In a blow up of the reflection, weak
 additional spots are observed, resulting from the long-range ordering of topological structures,
 with the periodicity of ~ 9 nm, consistent with Ref. 1. The reciprocal distance measured in the
 diffraction pattern is in agreement with the modulation observed in the real space image. **c**, GPA
 analysis of the STEM data reveals the in-plane strain ϵ_{xx} which exhibits a sinusoidal array.

Regarding the physical origin underlining the topological transformation of the domain patterns
 as a function of PTO thickness, we tried to explain the evolution from an energy point of view, in
 which the PTO energy density of various phases can reveal the phase transition sequence with
 the increase in the PTO thickness. As shown in Fig. R5 (supplementary Fig. S8), our phase field
 calculation results reveal that the decrease in the energy density of PTO layers is partially
 attributed to a drop of the average elastic energy density, which is reasonable since the flux-
 closure has the highest ratio of out-of-plane polarization, followed by the vortex state.
 Meanwhile, the electric and gradient energy density increase owing to the phase transition from a

wave-like state, to a smoothly rotating vortex state, and to a flux-closure state with more distinct
domain walls. Furthermore, the theoretical analysis by Z.H et al (Ref. 3) reveals a similar
tendency as our experimental results. We have added the calculation results to the supplementary
information in Fig. S8.

**Fig.R4. HAADF-STEM images with the corresponding a/c ratio of the lattice and strain**
**mappings. a**, HAADF-STEM image of a $(\text{PbTiO}_3)_n/(\text{SrTiO}_3)_{10}$ multilayer, with $n=19$ and $n=17$.
Boxes labeled as 1 and 2 denote representative areas. Box 3 is the reference lattice for GPA
method. **b**, Mapping of the a/c ratio in PTO and STO. **c** and **d**, Atomically resolved HAADF-
STEM images corresponding to the areas 1 and 2 in (a). The yellow arrows denote the
polarization direction. **e** and **f**, GPA analysis of the STEM data reveals that both in-plane and
out-of-plane strain exhibit a sinusoidal array, which is in accordance with a/c ratio.

**Fig.R5. The energies for PTO layers as a function of PTO thickness.** The evolution of energy
 components of ferroelectric domain reveals the phase transition sequence with increasing PTO
 thickness.

**Comment 3:** The appearance of electron carriers in ferroelectrics is fundamentally for screening
 the diverging electrostatic potential of relevant head-to-head dipoles. By contrast, the vortex core
 and the polar wave are formed by head-to-tail dipoles as depicted by the authors in Fig. 2, for
 which neither the elegant dipole closure of a vortex (Fig. 1d) nor the head-to-tail polar-wave
 intersection (Fig. 4) would require screening charges. As a result, the EELS interpretation of
 electron accumulations at the vortex core (Fig. 3) and the polar-wave junction (Fig. 4) becomes
 difficult to understand for this referee. As stated by the authors, the crystal-field splitting in the
 PTO is different from that in the STO. The direct correlation of a change in the crystal-field
 splitting to the presence of Ti³⁺ (thus electrons) in Fig. 3d is, therefore, unconvincing to this
 referee. The change in a crystal-field splitting and the presence of Ti³⁺ component can by far be
 also evident in the corresponding O K-edge spectra. Did the authors cross-check the Ti L- and O
 K-edge spectra? Furthermore, did the authors acquire the reference Ti L- and O K-edge spectra
 in both the bulk-PTO and -STO standards and compare them to the film counterparts? Is the
 conclusion drawn from Figs. 3 and 4 still robust? A quantitative estimation of a diverging
 potential and the screening charges required would be helpful for interpreting the EELS results.

**Response:** We totally agree that electron carriers in ferroelectrics are fundamental for screening
the diverging electrostatic potential of relevant head-to-head dipoles. While the trapped electrons
in vortex/wave layers are mainly distributed in core-region where polarization is near zero (as
shown in Fig. 3c), but not at the head-to-tail region as shown in Fig. R2.

Considering the origin of the electron concentrations: 1) for out-of-plane polarization, our
experimental evidence supports that it is driven by the demand of a screening mechanism as
discussed in the main text and in response to comment 1; 2) for vortex and wave layers, their
polarization transforms successively and exhibits a continuous rotation of the polarization, which
is different from the conventional heterojunction thin films. Correspondingly, the charge trapped
here could not be totally explained from convention aspects (such as charge compensation). One
possible origin is that the formation of oxygen vacancies at the vortex cores would attract
electrons. If we look at the local pressure distribution, the pressure at the vortex cores is negative
as shown in Fig.R6, and it is consistent with this hypothesis since oxygen vacancies have a larger
atomic volume than oxygen atoms. Our EELS measurements support this hypothesis as shown in
Fig. R7, oxygen vacancies are found to be distributed near the core-region. While the vortex
formation is such a complicated process which is influenced by a combination of competing
factors, an in-depth discussion on the origin of electron concentration call on further theoretical
work.

**Fig. R6 (a) Polarization distribution showing vortex arrays; (b) pressure distribution based**
**on phase field modeling.** The local pressure is equal to $-(\sigma_{11}+\sigma_{22}+\sigma_{33})/3$, where
σ_{11} , σ_{22} and σ_{33} are the local stress components, i.e. local pressure is equal to the
negative of average of diagonal stress components

For the correlation of energy splitting to the presence of Ti^{3+} : according to previous literature
 (Ref. 35), a typical Ti^{3+} EELS spectrum from Ti_2O_3 has two peaks of L3 and L2 edges. A narrow
 split as shown in Fig.R7b (blue curve) stands for a larger proportion of Ti^{3+} , which is in
 accordance with previous literature (Ref. 36,37).

As a cross certification, we measured the spectrum of the O K-edge as shown in Fig. R7. The
 peak located at ~ 537 eV (marked as peak 2) in the O K edge is closely related to the bonding
 state between oxygen and the surrounding cations (Ref. 40), and the enhanced peak 2 of the
 blue/green spectrum points toward the appearance of oxygen vacancies in the PTO layer (Ref.
 41), the presence of oxygen vacancies could provide electrons, which also supports our findings
 of Ti^{3+} .

The reference Ti L-2,3 and O-K spectra acquired in the bulk PTO has been added in the main
 text (Fig. 3d) and Supplementary materials (Fig. S11). We performed further measurements and
 ruled out the possibility of octahedral deformation inducing a crystal-field splitting as introduced
 in Supplementary Fig. S10. The situation is found to be similar in case of the core of polar-wave.
 For a more rigorous statement, we also updated the manuscript .

**Fig.R7. EELS characterization of a micro region in the vortex layer.** **a**, The upper colored
 surface plot shows the Ti-L3 energy splitting in the PTO layer where a vortex exists, the lower
 HAADF image shows the location of the corresponding vortex structure in the PTO layer. **b**, Ti-
 $L_{2,3}$ spectra corresponding to areas in (a). The grey curve: reference Ti- $L_{2,3}$ and O-K spectrum
 acquired in the bulk-PTO; the blue-A curve was acquired in the vortex core; the green-B curve
 was acquired in PTO layer but away from the vortex cores. **c**, EELS results of O-K edges
 acquired at different regions as mentioned in (a).

**Comment 4:** STEM-EELS is indeed capable of probing screening charges to ferroelectric
dipoles but unable for tackling the bound charges of given dipoles. In this regard, the authors'
interpretation of unveiling the ferroelectric-bound charges in a down-pointing film region (Figs.
4c-e) becomes puzzling. Again, comparisons to the bulk-PTO spectra are indispensable in order
to clarify the issue.

**Response:** We absolutely agree with the opinion of the referee. An efficient and precise way to
track the bound charges at the nanoscale still does not exist. Consequently, there is no way to
measure all charges for screening the bound charges in such localized region. However, it has
been a common practice to use the valence change of cations or oxygen vacancies to evaluate the
type and relative density of the screening charges (for example: Ref. 27 and Ref. 28). In our
previous works on the STO/PTO interface system, we have proved using both theory and
experiment [Advanced Materials 30 (38), 1707017; ACS applied materials & interfaces 10 (12),
10536-10542] that the ferroelectric polarization discontinuity at the interface leads to partially
occupied Ti 3d states. This means that the electrons are accumulated for polarization screening at
the positive polar interface of STO/PTO, thus a valence change of Ti can be observed from Ti^{4+}
to Ti^{3+} . In summary, we cannot detect all polarization-bound charges, but we do present solid
evidence that the electrons are accumulated at the particular interfaces and vortex cores.
Furthermore, the charged domain wall has quite similar features (Ref. 29), an n-type domain wall
conductivity, where electrons accumulate, associated with an Fe valence decrease. We have
rewritten the related paragraphs to clarify the discussions on charge accumulation at the vortex
cores.

A comparison to the bulk-PTO spectra has been added to main text and supplementary as Fig. 3d
and Fig. S11.

In summary, the structural characterization part of this work is decent. In comparison, the
electronic interpretation part is unsatisfactory and, in some places, contradicting to the
established wisdom. Indeed, this electronic part is essential for the titled subject and extensive
revisions would be required before further considerations of this manuscript.

**Response:** Following the referee's suggestion, we have updated some of the statements in our
manuscript, and a series of experimental and simulation results have been added as discussed

above. Combining the supplementary experiments, we hope to convince the referee on the charge
trapping at the vortex cores.

**Comment 5 (Minor point):** The ABF imaging does not require extremely thin specimens
according to this referee's hand-on experience. Therefore, it would be important for the authors
to be specific about the specimen-thickness requirements of the respective iDPC and ABF
techniques for the merit of potential interest parties.

**Response:** For this PTO/STO multilayer, usually a sample thickness of 50~60 nm is enough for
the iDPC technique to image oxygen atoms. While for the ABF technique applied to the same
sample, oxygen atoms could only be seen at the edge of a sample (the thinnest part). Ref. 18
compares both techniques under the same experimental conditions, iDPC clearly shows visible
atoms for GaN while in the ABF image they are difficult to be visualised. Besides, Ref. 17 points
out that the iDPC technique gives better precision than the ABF technique for measuring the
atomic column positions. Therefore, we chose iDPC rather than ABF to acquire the positions of
the oxygen atoms.

To clarify this, we modified the statement "ABF-STEM requires an extreme thin sample to
image light elements" in the main text.

Response to Referee #3

**General comment:** The manuscript reports atomic scale structure and evolution of topological
textures in a PbTiO₃/SrTiO₃ (PTO/STO) superlattice. By using the advanced TEM techniques
such as iDPC-STEM, EELS, and in-situ non-contact bias, as well as high-resolution STEM, the
authors successfully reveal the delicate ferroelectric structures emerging in the thickness-
controlled PTO sub-layers of the superlattice. The image quality looks excellent and convincing.
One of the main findings is observation of negative charge accumulation at the core of the
topological vortices and the EELS measurement results provide useful hints on the electronic
structures at the local areas. Since the topological textures in ferroelectrics are an emerging field
attracting a lot of attentions recently, the findings in a hot topological system have potential
merits to be published. But, the paper still has room for improvement and raises the following
questions and concerns as listed.

**Response:** We'd like to thank the referee for his/her careful reading, valuable comments, and
positive evaluation. We are delighted to see that Referee#3 speaks highly of our work. We took
the remarks of the referee very seriously and improved our manuscript accordingly. Below we
will respond to the specific comments and better clarify our results.

**Comment 1:** The correct topological terminologies should have used to describe the chiral
vortices. Vorticity is a different concept from chirality. In the caption of Fig. 1(c), the phrase of
"The red and blue regions indicate the vortex–antivortex pairs." is not correct. According to 2D
winding number calculation, two singular points in the red and blue regions appear to have a
topological charge of +1, respectively. And thus, they both should be called vortices rather than
using the antivortex. Since they can be distinguished in aspect of chirality, "clockwise or
counterclockwise chiral vortices" are topologically meaningful expressions to indicate the two
regions. More rigorous description of the topological charges in ferroelectrics can be found in
recently published references [Kim, K. E. et al. Nature Communications 9, 403 (2018), Li, Y. et
al., npj Quantum Materials 2, 43 (2017), Seidel, J. (Ed.) Topological structures in ferroic
materials. (Springer, 2016)], which are necessary to be cited in addressing the topological
structures.

**Response:** We'd like to thank the referee for bringing this up. Comparing with “antivortex”
raised in Ref [Kim, K. E. et al. Nature Communications 9, 403 (2018)], [Li, Y. et al., npj
Quantum Materials 2, 43 (2017)], we agree that “clockwise or counterclockwise vortices” (We
are not sure if they are chiral) is indeed a more accurate description for the topological structure
mentioned in this article.

We have updated our manuscript accordingly and also cited the references as Ref. 19, Ref. 20
and Ref. 21.

**Comment 2:** Specify where the biased voltage is applied during the in-situ bias experiment.
Regardless of the explanation in the Methods section, the experimental geometry of non-contact
electric bias experiment is still unclear. Is it possible to quantify the electric field across the
corresponding layer of PTO?

**Response:** The experimental geometry of the electric bias experiment is shown as Fig. R8.
Actually, we also tried the contact bias mode, in which the needle directly contacts with the top
of multilayer at the initial state. The shaking needle brought enormous difficulties to the subtle
STEM experiment, and the lamella is easily broken in this mode.

For the non-contact bias mode, the distance between the tungsten probe and the lamella can be
technically controlled to be less than 3 nm, according to our hand-on experience, when applying
a bias voltage, the lamella will be attracted to the tungsten probe, and the final distance between
them will be shortened to even less. The thickness of the multilayer used for the in-situ bias
experiment is ~60 nm. Although in our experiment the situation is complex (the irregular shape
of the tungsten probe and the lamella will influence the precision of the evaluation), using the
dielectric constant given in [Nature 565, 468–471 (2019)], we carried out phase-field simulations
and the out-of-plane component of the electric field in the PTO layer is up to hundreds of
kilovolt per centimeter and homogenous due to formation of a uniformly polarized state to align
the polarization in the electric field direction.

[Redacted]

**Fig.R8. a**, STEM image describing electric bias experiment and the relative position of every
 section. **b**, the dielectric constant of the superlattice and the STO thin film. Cited from [Nature
 565, 468–471 (2019)].

**Comment 3:** In Figs. 2b-d, the white dots indicating negative charges are displayed on the
 experimentally obtained polarization maps. Is this charge distribution also experimentally
 confirmed or just guessed from polarization discontinuity? As in the Fig. 2d, one can easily
 imagine the bottom interface of polarization down region has a negative bound charge density.
 But, it would not seem simple to predict the singularities have negative charges in the vortex and
 wave configurations. The Figs. 2b-d have three columnar sub-panels consisting of two
 experiments and one simulation. What does the first column in the experimental part mean?

**Response:** As for the first question, we did measure the EELS information in the PTO layer with
 different thickness which contains different topological structures to detect the electron
 distribution. It is a common practice to use the valence change of the cation or oxygen vacancy
 to evaluate the type and relative density of the charges (for example: Ref. 27, 28, 29). In our
 previous works for the STO/PTO interface system, we showed using both theory and experiment
 [Advanced Materials 30 (38), 1707017; ACS applied materials & interfaces 10 (12), 10536-
 10542] that the electrons may partially occupy Ti 3d states, thus a valence change of Ti can be
 observed from Ti^{4+} to Ti^{3+} . Based on this method, we measured the O-K and Ti-L_{2,3} edges at the
 out-of-plane polarization region (as shown in main text Fig. 4c-e), the vortex structure (as shown

in main text Fig. 3d and supplementary Fig. S11) and wave structure (as shown in main text Fig.
4a-b).

To map the polarization in the whole 10-uc-PTO region (second question), the STEM images
would include too many arrows in our initial version. For the sake of readability, we added some
schematic diagrams as shown in the first column to make the corresponding topological structure
visually clearer. Following the referee's comment, we updated the figure caption and modified
the annotation of first column into a "schematic diagram".

**Comment 4:** In the following EELS experiments, the t_{2g} - e_g splitting in the L3 or L2 peaks is
used as an indicator of electron doping. As argued by the authors, the decrease in the splitting
can be attributed to the existence of Ti^{3+} along with Ti^{4+} . But, this may not be the unique
reason. Basically, the t_{2g} - e_g level splitting is due to the crystal field from the neighboring anions.
The vortex areas have non-collinear arrangements of polarizations, which can result in complex
octahedral deformations. Exact electronic structure in the situation is not disclosed yet. Without
exclusion of the possible concern or minimal warning, such innocent interpretation can fall into
error.

**Response:** We totally agree with the referee's opinion that complex octahedral deformations
may lead to the t_{2g} - e_g level splitting as well. To evaluate the magnitude of octahedral deformation,
we tried to observe and calculate the octahedral deformation based on our best iDPC images as
shown in Fig.R9 and compare different regions (*e.g.* between inside-vortex-core and outside-
vortex-core).

The oxygen octahedral distortion/tilt and displacements in perovskite oxides are known to play a
significant role in determining the functionalities like magnetic, electric and ferroelectric
polarization [Jia, C. L. et al. Nat. Mater.7, 57–61(2008); Bousquet, E. et al. Nature 452, 732–736.
(2008)]. Particularly, Balke, N. et al. did DFT calculations of the electronic structure at the
vortex inside BFO, in which the octahedral tilt is believed to be able to induce potential changes
in the electronic structure (Ref. 38). Based on a model-based method for quantitative electron
microscopy in which images are modelled as a superposition of 2D Gaussian peaks [De Backer,
394 A., Ultramicroscopy 171, 104-116 (2016)], we determined the octahedral tilt based on iDPC
images and the results are shown in Fig .R9. From the statistical results (Fig. R9a), we found tilt

angles below 1.5° in most unit cells of the PTO layer, which is significantly lower than the angle
 which is assumed to induce significant changes in the electronic structure (Ref. 38). Moreover,
 when comparing the tilt angles in the vortex core and in non-core regions, no significant
 difference was observed as shown in Fig. R9b.

 **Fig. R9. Analysis of the oxygen octahedral tilt based on iDPC images.** (a) Statistical results of
 the oxygen octahedral tilt angle. (b) Distribution of the octahedral tilt across the vortex core
 region. (c) the iDPC-STEM image used to calculate the octahedral tilt.

We further checked the elongation and rotation of the oxygen octahedra, and found that both of
 them are limited to minimal degrees, and no obvious distinction between vortex core and other
 regions is seen. The wave structure reveals the same results.

In summary, we believe the octahedral deformation and local distortion are not the main reason
 for such huge t_{2g} - e_g splitting change at the core of vortex, the most possible explanation is the
 accumulation of electrons. And then the excess electrons partially occupy the Ti 3d states. We

have updated related descriptions in main text, the possibility of other factors inducing the
electronic structure is mentioned.

**Comment 5:** How can the authors quantify the ratio of Ti³⁺ and Ti⁴⁺ concentrations in Fig. 4e?
Is it proportional to the ratio of integrated intensities of t_{2g} peak relative to the e_g peak because
the Ti³⁺ partially occupies the t_{2g} orbitals? If so, the physical meaning of the spectroscopic
results should be more kindly and strictly described in the main text related to the figure.

**Response:** For the first question, we used a model based quantification of the EELS spectra to
estimate the relative concentrations of Ti⁴⁺ and Ti³⁺, in which the electron energy loss spectra
results were fitted based on standard Ti³⁺ and Ti⁴⁺ spectra and the proportion of each valence
state was given. This method and quantification software are based on Ref. 30.

For the second question, the interpretation of the EELS fine structure is not always
straightforward, since the intensity and feature reflect the unfilled density of states (DOS) above
the Fermi energy level, which in turn is sensitive to the bonding and the valence state. A typical
Ti³⁺ EELS spectra from Ti₂O₃ has two peaks of L₃ and L₂ edges, as shown in Fig. R10a (Ref.
35). The feature of decreasing splitting between t_{2g} and e_g doesn't mean the Ti³⁺ partially
occupies the t_{2g} orbitals, simply reflects that the linear component of Ti³⁺ is getting increased as
shown in Fig. R10b (Ref. 34-37).

Following the referee's suggestion, we have updated some statements on the EELS results in the
main text.

[Redacted]

**Fig. R10. a**, Reference spectra for Ti^{4+} (red) and Ti^{3+} (blue) are shown, taken from thick sections
of SrTiO_3 and LaTiO_3 , and the spectra of mixed valence. **b**, Ti-L_{2,3} absorption spectra of Nd_{1-x}
TiO_3 as a function of vacancy compositions (x).

List of main changes

In general, we rewrote the texts discussing the electron distributions with added series of
EELS results and additional theoretical work. According to the comments of referees, some
terminologies and presentations are also improved. We believe we have improved the readability
of the manuscript.

- 1. The title was changed to “Manipulating topological transformations of polar structures
through real-time observation of dynamic polarization evolution”.
- 2. In abstract paragraph, “the confined charge in the core of polar vortex is found and
expected to play an important role in the process of topological structure transformation”
was modified to “Furthermore, the redistribution of charge in various topological
structures has been demonstrated under an external bias.” for more rigorous statement.
The similar expressions following are also modified for the same reason.
- 3. In introduction paragraph, “Using a combination of scanning transmission electron
microscopy (STEM) and in-situ non-contact bias technique, a real time mapping of the
ferroelectric polarization under an electric bias was carried out on an atomic scale. The
electron energy loss spectroscopy (EELS) results provide hints on the variation of the
electronic structure during the topological transformations. Our methods allow us to
realize a controllable transformation, and the connection of various topological structures
is experimentally presented” were added.
- 4. The statement “Unlike traditional techniques like ABF-STEM, this new technique does
not require an extreme thin sample to image light elements” was replaced by “Compared
with annular bright-field (ABF) STEM, iDPC provides better precision for measuring
the atomic column positions¹⁸” according to referee 2, comment 5.
- 5. In Fig. 1 caption, the terminology “vortex–antivortex pairs” was replaced by “clockwise
and counterclockwise vortex pairs”, according to referee 1, comment 1.
- 6. In Fig. 2 caption, the statement “The three columnar sub-panels are schematic diagrams,
mappings of polar vector and phase field simulation results, respectively” was added to
make the components of figure more understandable.
- 7. The strain-related paragraph was updated with further analysis as shown in
supplementary Fig. S5 and S7. Theoretical work on thickness-changed layer was added
as Fig. S8. A detailed introduction to GPA was added to method part.

- 8. The EELS interpretation part was rearranged. We strengthened the explanation of energy
splitting and further analysis on octahedral deformation was added (supplementary Fig.
S10). The demonstration of oxygen vacancies was added and detailed introduced (Fig.
4c-e and supplementary Fig. S11). Correspondingly, the figure caption was undated. The
screening mechanism within out-of-plane polarization region realized by Ti^{3+} and
oxygen vacancies was demonstrated and updated.
- 9. In “discussion and summary” part, the statement “the presence of Ti^{3+} and oxygen
vacancies provides a cross-check that shows a high correlation with the various
topological structures. These experimental evidences allow us to propose the possibility
of a one-to-one correlation between the charge and various domain configurations in this
oxide multilayer. Our findings support that in the out-of-plane polarization layer, the
electrons concentrated at the positive polar interface may be attributed to the polar
screening. While in the vortex, zig-zag vortex state, and the polar wave structure, where
polar screening is not needed, other factors such as the gradient of the electric field or the
local pressure may make a contribution. An analogous phenomenon was found and
explained in the field of optical vortices^{45,46}: optical tweezers are able to trap and
manipulate small particles, typically of micron size. Another possible origination is that
the formation of oxygen vacancies at the vortex cores would attract electrons. The local
pressure at the vortex cores is negative according to our phase field modeling. Oxygen
vacancies are favored to distribute near the core-region, since oxygen vacancies have a
larger atomic volume than oxygen atoms.” was added to replace the initial expression for
a more precise statement. An analogous phenomenon in optical field was introduced.
- 10. We acquired the reference spectrum of Ti and O in bulk PTO and added them in Fig. 3d
and supplementary Fig. S11.
- 11. New references Ref. 18-22, 26-29, 34-35, 37, 39-42, 45-46, 48-49 were added in the text,
according to referee 2: comment 1,3,4,5; referee 3: comment 1,4,5.
- 12. All changes in the main text have been highlighted in red.

Reviewers' comments:

Reviewer #2 (Remarks to the Author):

This referee deeply appreciates the tremendous experimental and theoretical efforts put by the authors to improve the work. The rewriting of the manuscript, however, brought up several self-inconsistencies (elucidated below), which are inadequate for an active consideration of the work in its present form.

1. As indicated by the new title of the manuscript, the most important exploration of the work is the rich evolution of the domain patterns upon increasing electric biases (Fig. 2). In this revision, the authors raised the issue of an electron concentration at the vortex core, assisted by the oxygen-vacancy formation. Hence, the field-dependent domain evolution is also to be entangled with the field-driven redistribution of oxygen vacancies, as suggested in Figs. R1 and R2 though not thus argued. An electric-field driven redistribution of oxygen vacancies has been a well-known phenomenon, while this is not the concern of this referee. The profound concern of this reviewer is rather that the energetics proposed in Fig. R5 would then need to incorporate this additional term, not to mention that none of each contributing term in Fig. R5 has been clearly defined in the Methods section. More seriously, Fig. R5 and the corresponding Fig. S8 show totally different energy-density scales. The physics behind Fig. R5 and the corresponding Fig. S8 readily become questionable and all related paragraphs added to the revised manuscript are equally problematic.

2. Significant efforts have been paid to the spectral interpretation of Ti-L and O-K edges. However, the authors' argument on Ti³⁺ throughout the manuscript should be accompanied with an estimated ratio from the corresponding spectral fitting. None of such a quantitative estimation of the Ti³⁺ fraction has been described in the work. Take Fig. 3d for example, if the Ti³⁺ ratio is indeed large or larger as suggested by the authors, the t_{2g}-e_g splitting shall then be washed out considering the characteristic Ti L-edge spectrum of Ti³⁺ does not show the t_{2g}-e_g splitting. However, Fig. 3d does not reveal such a corresponding spectral signature.

3. What is the EELS detection limit of the authors' apparatus? On page 11 (line 187), the discussed "high concentration of electrons (~10²⁰ cm⁻³) or oxygen vacancies" would nonetheless correspond to a detection limit below 1 atomic percent, the probing of which is unlikely even for state-of-the-art EELS facility, and renders the spectral argument on oxygen vacancies largely uncertain. In PTO, an oxygen vacancy can pair up with a lead vacancy, casting an overall charge neutrality. This referee is conservative about the authors' argument on an electron doping by oxygen vacancies considering also that the EELS spectra in Figs. 3 and 4 are indeed too noisy (summing up more spectra of a comparable quality can resolve the problem). It is, however, true that an oxygen vacancy can indeed lead to a lattice dilation and should then lead to a positive pressure around the vortex cores (Fig. R6). Why it is indicated as a negative pressure in Fig. R6?

4. Many typos appear in this revision package.

It is this referee's general impression that the authors may have rushed to the writing of the revised manuscript after extensive additional experimental and theoretical tasks, thus leading to the above inconsistencies that are not supposed to appear at the present stage. Indeed, this is a good work from the structural-characterization aspect, while the corresponding electronic part still does not reach the minimum requirement of a self-consistency. This referee could not recommend the publication of the work at this moment.

Reviewer #3 (Remarks to the Author):

Since the questions are reasonably answered and the requests are reflected in the revised manuscript, I would like to recommend acceptance for publication in Nature Communications.

Point-by-Point Response to Referees

Response to Referee #2

General comment: This referee deeply appreciates the tremendous experimental and theoretical efforts put by the authors to improve the work. The rewriting of the manuscript, however, brought up several self-inconsistencies (elucidated below), which are inadequate for an active consideration of the work in its present form.

Response: We thank the referee for the positive comments about the new results we provided in the revised paper. While after carefully checking the manuscript, we agree with that some description may be not clear or accurate enough, but respectfully disagree with the comment of “self-inconsistencies”. The point-by-point response to the referee’s suggestions is in the following, the points which may mislead the referee are clarified one by one.

Comment 1: As indicated by the new title of the manuscript, the most important exploration of the work is the rich evolution of the domain patterns upon increasing electric biases (Fig. 2). In this revision, the authors raised the issue of an electron concentration at the vortex core, assisted by the oxygen-vacancy formation. Hence, the field-dependent domain evolution is also to be entangled with the field-driven redistribution of oxygen vacancies, as suggested in Figs. R1 and R2 though not thus argued. An electric-field driven redistribution of oxygen vacancies has been a well-known phenomenon, while this is not the concern of this referee. The profound concern of this reviewer is rather that the energetics proposed in Fig. R5 would then need to incorporate this additional term, not to mention that none of each contributing term in Fig. R5 has been clearly defined in the Methods section. More seriously, Fig. R5 and the corresponding Fig. S8 show totally different energy-density scales. The physics behind Fig. R5 and the corresponding Fig. S8 readily become questionable and all related paragraphs added to the revised manuscript are equally problematic.

Response: We thank the referee for his/her careful reading of our response. Our experimental evidences point towards the existence of oxygen vacancies which could give a reasonable explanation for the Ti valence changes, and they play a key role in the polarization screening mechanism.

Actually, the detailed definition of each contributing term in the energy evolution graph has been raised in the Methods section in the form of references (Ref. 51-56). We are glad to put those equations into the method part, the expression for each energy density have been added.

The evolution of energy components is presented to reveal the phase transition sequence with the increase in the PTO thickness from an energy aspect. We apologize for providing two graphs about energy evolution which misled the referee. These two graphs are both from phase field modeling, the result shown in Fig. S8 is set up with a more accurate dielectric constant in the calculation initial conditions, and its corresponding calculated domain morphology fit our experimental results in a better degree (thus the results shown in the supplementary materials Fig. S8 should be preserved). We mistakenly put both of them in the previous revision package, while this does not change any discussion or summary of this part, just showing the same trend of energy evolution with different initial conditions.

We are happy to give a more detailed introduction here to explain the physical meaning underlying this energy evolution graph as shown in Fig. S8, besides the figure caption mentioned in the supplementary materials. According to [Damodaran, A. R., et al. *Nat. Mater.* 16,1003-1009 (2017)], the analysis of energy change is effective to describe the phase transition. The decrease of the average elastic energy density could be understood since the vortex state has a higher ratio of elastically favorable out-of-plane polarization, and the flux-closure state has the highest ratio of out-of-plane polarization [Hsu S. L. et al. *Adv. Mater.*, 1901014, (2019)]. The landau energy density decreases owing to increasing out-of-plane polarization. Meanwhile, electric and gradient energy density increase owing to the phase transition from the wave-like state, to rotational vortex state smoothly, and to flux-closure state gradually with more distinct domain wall [Hong, Z.,

et al. *Nano Lett.* 17, 2246-2252 (2017)]. From an energetic point of view, the film thickness-driven phase transition is a result of the competition between the individual energies—elastic, electric, Landau, and gradient. More details about this theoretical part about the domain evolution and corresponding energy evolution could be found in Ref. 3 [*Nano Lett.* 17, 2246-2252 (2017)] and Ref [Adv. Mater., 1901014, (2019)].

Comment 2: Significant efforts have been paid to the spectral interpretation of Ti-L and O-K edges. However, the authors' argument on Ti³⁺ throughout the manuscript should be accompanied with an estimated ratio from the corresponding spectral fitting. None of such a quantitative estimation of the Ti³⁺ fraction has been described in the work. Take Fig. 3d for example, if the Ti³⁺ ratio is indeed large or larger as suggested by the authors, the t_{2g}-e_g splitting shall then be washed out considering the characteristic Ti L-edge spectrum of Ti³⁺ does not show the t_{2g}-e_g splitting. However, Fig. 3d does not reveal such a corresponding spectral signature.

Response: We thank the referee for reminding us of this point. We totally agree that it's important to present an estimated ratio. Actually, we have given such a quantitative estimation of Ti³⁺ fraction in the main text and supplementary materials. Using model-based quantification of the EELS spectra, the relative concentrations of Ti⁴⁺ and Ti³⁺ were estimated.

Such as:

- 1) Fig. 3c is the superposition of the Ti⁴⁺ and Ti³⁺ signal based on EELS analysis, color here represents the distribution and fraction contribution of Ti⁴⁺ and Ti³⁺ component.
- 2) Fig. S13c shows the Ti³⁺/Ti⁴⁺ ratio in the PTO and STO layers across the multilayer. The value of Ti³⁺/Ti⁴⁺ ratio is given on the left vertical coordinate axis.

According to Ref. [Y. Shao et al. *Ultramicroscopy* 110, 1014-1019 (2010)], the t_{2g}-e_g splitting is still distinct when the Ti³⁺ fraction is ~ **0.249** (x=0.10) as shown in Fig. R1a. The hint of splitting could be observed even at a higher Ti³⁺ fraction of ~ 0.536 (x=0.20). According to our estimated Ti³⁺ fraction in the vortex layer shown in Fig. R1b, the average value of Ti³⁺ fraction in core-region is ~ **0.19**, which means the t_{2g}-e_g splitting

should be quite clear rather than “be washed out”. Comparing our spectrum with the reference spectrums of $x=0.10$ and $x=0.02$, a high degree of consistency could be found. For the out-of-plane polarization as shown in Fig. S13a, the Ti^{3+} fraction would be less than 0.5 (for 8-uc PTO layer) based on our estimation as shown in Fig. S13c, compared with that in Fig. R1 ($x=0.20$, Ti^{3+} fraction is ~ 0.536), the Ti edge would contain a clear L3-splitting and an obscure L2-splitting, which also shows a high degree of consistency with our experimental results as presented in Fig. S13b (the bottom spectrum).

[Redacted]

Fig. R1. **a**, Ti-L_{2,3} edge from the BaTi_{1-x}Nb_xO₃ polycrystalline samples and BaTi_{0.998}Nb_{0.002}O₃ (Ti⁴⁺) and BaTi_{0.5}Nb_{0.5}O₃ (Ti³⁺). The black circles are the experimental spectra and the red solid lines are the fitted spectra using the reference spectra at the top and bottom of the series with a normalization factor. This is cited from [Y. Shao et al. *Ultramicroscopy* 110, 1014-1019 (2010)]. **b** and **c**, The estimated Ti³⁺ fraction and corresponding HAADF image, the quantitative estimation is based on EELS data which is acquired in the green region. The method and quantification software are based on Ref. 30 [*Ultramicroscopy* 101, 207-224 (2004)].

Actually, there are the other two common methods which are widely used to measure the Ti valence changes: the L_{2,3} intensity ratio and the energy splitting; while the L_{2,3}

intensity ratio has some shortcomings for high energy resolution data as discussed in Ref. [*Ultramicroscopy* 110, 1014-1019 (2010)], thus it is the more suitable choice to use the t_{2g} - e_g splitting to measure the proportion of the Ti^{4+} and Ti^{3+} components as used in our work.

Meanwhile, the high-end EELS equipment with a monochromator was used in our work, in which the energy resolution could be significantly improved to ~ 0.3 eV as shown in Fig. R2 (Generally, the energy resolution of EELS data is ~ 1.0 eV in many other literatures), thus we have the opportunity to observe the extremely detailed characteristic splitting.

Fig. R2. The zero-loss peak acquired during our experiment, the full width at half maximum (FWHM) is ~ 0.3 eV under the dispersion of 0.1 eV/channel.

In the revised main text, we have added “According to the model-based quantification, the average Ti^{3+} fraction in a typical core-region is calculated to be ~ 0.19 ” besides the mapping of Ti valence fraction in Fig. 3c.

Comment 3: What is the EELS detection limit of the authors’ apparatus? On page 11 (line 187), the discussed “high concentration of electrons ($\sim 10^{20}$ cm $^{-3}$) or oxygen vacancies” would nonetheless correspond to a detection limit below 1 atomic percent, the probing of which is unlikely even for state-of-the-art EELS facility, and renders the spectral argument on oxygen vacancies largely uncertain.

Response: We thank the referee for his/her careful reading.

I. I. Ivanchik raised that the electrons/vacancies could accumulate and play a role in screening in the **bulk density** of the order of 10^{20} cm^{-3} in Ref. [*Ferroelectrics*, 145(1), 149-161 (1993)]. While considering that in single domain, most of the electrons/vacancies concentrated in the extremely **narrow surface/interface layer**, the charge density in this local area would be rather higher than 10^{20} cm^{-3} . Taking the reported bulk density value of $5 \cdot 10^{20} \text{ cm}^{-3}$ in BaTiO_3 for an example, with the domain thickness of 8 unit cell, under ideal conditions in which most the electrons are concentrated in the surface layer, a total of ~ 0.25 electrons per unit cell in the surface layer (namely $\sim 4\%$ atomic percent of oxygen vacancies) could take part in the ferroelectric screening. According to Muller, D. A. et al. [*Nature* 430, 657-661(2004)], $\sim 1\%$ oxygen vacancies could be detected through EELS measurement. Thus, it is reasonable to detect the oxygen vacancies.

According to R. V. Wang et al. [*Phys. Rev. Lett.* 102, 047601 (2009)], the oxygen vacancies could exist in the polar surface to compensate for the polarization in PbTiO_3 film, and the **surface charge density** is ~ 0.5 electrons per unit cell (namely 8% atomic percent of oxygen vacancies). This value is very close to our experiment results as shown in Fig. S13.

To make our claim more accurate, we have added the statement of “with bulk density of the order of” in front of $\sim 10^{20} \text{ cm}^{-3}$ in the main text.

In PTO, an oxygen vacancy can pair up with a lead vacancy, casting an overall charge neutrality. This referee is conservative about the authors' argument on an electron doping by oxygen vacancies considering also that the EELS spectra in Figs. 3 and 4 are indeed too noisy (summing up more spectra of a comparable quality can resolve the problem). It is, however, true that an oxygen vacancy can indeed lead to a lattice dilation and should then lead to a positive pressure around the vortex cores (Fig. R6). Why it is indicated as a negative pressure in Fig. R6?

Response: We thank the referee for pointing this out. An overall charge neutrality is reasonable for entire bulk, while taking both the accumulated electrons at topological structures and the oxygen vacancies into consideration, it does not against the principle

of overall charge neutrality. Besides, according to Ref. 42 [*ACS Appl. Mater. Interfaces* 6, 11980-11987. (2014)], the Pb vacancies and O vacancies would lead to totally opposite changes in O-K edge: the second peak will be noticeably weakened by the Pb vacancies but strengthened by the O vacancies, as shown in Fig. R3. In fact, it's not rare in other literatures that the oxygen vacancies could provide electrons for screening and this claim has been an established wisdom. [Such as Ref. 27: *Nat. Mater.* 13, 1019-1025 (2014); Ref. 28: *Nat. Commun.* 9, 1638 (2018); Ref. 39: Fridkin, V. M. *Ferroelectric semiconductors* (Plenum Press, New York, 1980); Ref. 43: *Phys. Rev. Lett.* 102, 047601 (2009); and *Adv. Mater.*, 30, 1707017(2018)]

[Redacted]

Fig. R3. Calculated O K edges via the first principle modeling [cited from Ref. *ACS Appl. Mater. Interfaces* 6, 11980-11987. (2014)]. The first principle modeling analysis results confirm that the Pb vacancies and O vacancies would lead to totally opposite changes in the second peak of O-K edge.

For the EELS spectrum presented in Fig. 3 and Fig.4, they have already been processed through summing up aimed at reducing the noise. While the EELS spectrum could not be as smooth as the optical spectrum (such as XAS) due to the limit of the instrument and beam sensitivity of the sample to a greater or lesser extent. Increasing acquiring time and dose could further promote the spectrum quality while this would cause larger beam damage to the sample, in which the crystal lattice would be destroyed and severe carbon deposition would happen. Different experiment parameters had been considered

and compared when we acquired these data. For example, the EELS data in this article required a high energy resolution, thus the monochromated EELS equipment was used, which would reduce the spatial resolution, we have to seek the balance between these parameters and we have tried our best to increase the spectrum quality through many methods.

We totally agree with your saying that the lattice dilation and positive pressure are led by oxygen vacancies. Actually, we are talking about the same phenomena. Here, we use the hydrostatics pressure with the form: $-(\sigma_{11}+\sigma_{22}+\sigma_{33})/3$, where σ_{11} , σ_{22} and σ_{33} are the local stress components. The hydrostatics pressure, thus, can be considered as the reaction force of the pressure that vortices are given.

The vortex cores show the negative hydrostatics pressure (namely the positive pressure) both in experiments (Fig. S6) and simulations [Hong, Z.J. Phase-Field Simulations of Topological Structures and Topological Phase Transitions in Ferroelectric Oxide Heterostructures. (Doctoral dissertation, 2017)], and it is in consistent with the accumulated oxygen vacancies.

Comment 4: Many typos appear in this revision package.

Response: According to the referee's advice, we have carefully checked all the submitted files again, and the discovered typos have been corrected.

Such as, in line 14 the "offer" have been corrected to "offers", in line 94 the "a" have been added, and so on.

It is this referee's general impression that the authors may have rushed to the writing of the revised manuscript after extensive additional experimental and theoretical tasks, thus leading to the above inconsistencies that are not supposed to appear at the present stage. Indeed, this is a good work from the structural-characterization aspect, while the corresponding electronic part still does not reach the minimum requirement of a self-

consistency. This referee could not recommend the publication of the work at this moment.

Response: We sincerely thank the referee for his/her positive view on our work from the structural-characterization aspect, indeed the main focus of this article is using high-end structural-characterization methods to explore the topological structures.

With the added theoretical and experimental work above, we believed the referee's concerns have been well addressed. As quoted from the other two referees, our present work is of high quality and have the original achievements.

Referee #1: "...the electric field control of the vortices and associated changes in chemical structure (EELS) in my opinion make this an interesting and original study..."

Referee #3: "...The image quality looks excellent and convincing...the topological textures in ferroelectrics are an emerging field attracting a lot of attentions recently..."

We hope now the referee can be satisfied with our revised manuscript and agrees with the publication in Nature Communications

Response to Referee #3

Comment: Since the questions are reasonably answered and the requests are reflected in the revised manuscript, I would like to recommend acceptance for publication in Nature Communications.

Response: We thank the referee for his/her recommendation for publication. His/her valuable advice had a profound effect on improving the quality of our paper.

REVIEWERS' COMMENTS:

Reviewer #2 (Remarks to the Author):

The concerns of this referee are now largely resolved and this reviewer would like to suggest the publication of the work.